# Impacts of Habitual Diets Intake on Gut Microbial Counts in Healthy Japanese Adults

**DOI:** 10.3390/nu12082414

**Published:** 2020-08-12

**Authors:** Takuya Sugimoto, Tatsuichiro Shima, Ryuta Amamoto, Chiaki Kaga, Yukiko Kado, Osamu Watanabe, Junko Shiinoki, Kaoru Iwazaki, Hiroko Shigemura, Hirokazu Tsuji, Satoshi Matsumoto

**Affiliations:** 1Yakult Central Institute, 5-11 Izumi, Kunitachi-shi, Tokyo 186-8650, Japan; tatsuichiro-shima@yakult.co.jp (T.S.); ryuta-amamoto@yakult.co.jp (R.A.); chiaki-kaga@yakult.co.jp (C.K.); yukiko-kado@yakult.co.jp (Y.K.); hirokazu-tsuji@yakult.co.jp (H.T.); satoshi-matsumoto@yakult.co.jp (S.M.); 2Development Department, Yakult Honsha Co., Ltd., 1-10-30 Kaigan, Minato-ku, Tokyo 105-8660, Japan; osamu-watanabe@yakult.co.jp (O.W.); junko-shiinoki@yakult.co.jp (J.S.); 3Corporate Planning Office, Yakult Honsha Co., Ltd., 1-10-30 Kaigan, Minato-ku, Tokyo 105-8660, Japan; kaoru-iwazaki@yakult.co.jp (K.I.); hiroko-shigemura@yakult.co.jp (H.S.)

**Keywords:** gut microbiota, habitual diets, gut bacterial counts, healthy population, Japanese

## Abstract

Although diet is an important factor influencing gut microbiota, there are very few studies regarding that relationship in Japanese people. Here, we analyzed the relationship between habitual dietary intake surveyed by food frequency questionnaire and the quantitative features of gut bacteria by quantitative PCR and next generation sequencer in 354 healthy Japanese adults. The α-diversity of gut microbiota was positively correlated with the intake of mushrooms and beans and negatively correlated with the intake of grains. The β-diversity was significantly associated with the intake of fruits, mushrooms, seaweeds, seafoods, and alcoholic beverages. Multiple linear regression analysis of the relationship between food groups associated with the diversity of gut microbiota and the number of gut bacteria at the genus level found 24 significant associations, including a positive association between alcoholic beverages and the number of *Fusobacterium*. These results support that habitual dietary intake influenced the diversity of gut microbiota and was strongly associated with the number of specific gut bacteria. These results will help us to understand the complex relationship between habitual diet and gut microbiota of the Japanese.

## 1. Introduction

The human gut microbiota is diverse, with more than 100 × 10^12^ microorganisms in our body [1]. It is involved in a variety of physiological functions beneficial to the host, such as maintenance of intestinal epithelial cell homeostasis [2,3,4], protection against infection [5], and regulation of gut immune system [3,6].

The gut microbiota is also diverse among individuals and is influenced by age, sex, body mass index (BMI), lifestyle (such as smoking and exercise), antibiotic use, and diseases [7]. Food components are not only a nutrient source for gut bacteria, but some of them are also metabolized by gut bacteria to metabolites, such as short-chain fatty acids. Therefore, diet is considered to be an important factor influencing the gut microbiota [8]. Long-term dietary habits strongly affect individual enterotypes. For instance, humans with the Bacteroides enterotype consume much animal protein and fat on a daily basis; those with the Prevotella enterotype consume much carbohydrate [9]. The gut microbiota is also altered by the intervention of various food components, including resistant polysaccharides such as fructans and galacto-oligosaccharides [10], animal proteins such as meat, eggs, and cheese [11], and a high-fat diet [12]. However, few studies have analyzed the relationship between usual dietary intake and gut microbiota in healthy adults. A recent large-scale observational study of 862 French adults showed a positive association between usual intake frequency of fruits and fish and the α-diversity of gut microbiota, and a negative association between that of cheese and the abundance of *Akkermansia muciniphila* [13]. However, as major dietary components differ according to regions where people live, and as the gut microbiota is influenced by individual genetic background, the relationship between diet and gut microbiota needs to be investigated by region.

There have been very few studies of the relationship between diet and gut microbiota in Japanese people. Seura et al. surveyed diet for three consecutive days in 28 women in their twenties and analyzed their gut microbiota by using the terminal restriction-fragment-length polymorphism (T-RFLP) method; they reported a negative association between the intake of eggs and the abundance of *Clostridium* [14]. However, the age and sex of the subjects were biased and gut microbiota was not analyzed in detail. Therefore, we thought that a large-scale, detailed analysis of the relationship between diet and gut microbiota in healthy Japanese with a wide age range would enable us not only to clarify the association between diet and gut microbiota in Japanese people. In this study, we analyzed the relationship between habitual dietary intake surveyed using a food frequency questionnaire (FFQ) and gut microbiota analyzed quantitatively in 354 healthy adults, with the aim of revealing the association between habitual diet and gut microbiota.

## 2. Materials and Methods

### 2.1. Cohorts

We recruited 383 healthy Japanese adults from 20 to 60 years of age by KSO Co., Ltd. (Tokyo, Japan), in almost equal numbers by sex and age group [15]. This study was conducted in conformity with the Helsinki Declaration, and the study plan was approved by the Human Study Ethics Committee of Nihonbashi Cardiology Clinic (Tokyo, Japan) (Approval number NJI-017-09-01, date of acceptance 14 September 2017). All subjects gave their informed consent for inclusion before they participated in this study. Of the 383 subjects recruited, we excluded 17 who met the exclusion criteria (administration of antibiotics, antiflatulents, or laxatives within 1 week before stool collection) and 12 for whom insufficient 16S rRNA gene reads were detected in following 16S rRNA gene amplicon sequencing. The remaining 354 subjects were included in the analysis.

### 2.2. Collection of Dietary Data

We used a self-administered, semiquantitative FFQ developed by the National Cancer Center Japan to investigate the average daily dietary intake [16]. The subjects recorded the average frequency of consumption and the intake per meal of 155 food items over the past year in the FFQ. The data were tallied and the total daily energy consumption and average daily intake of 13 food groups (grains, potatoes, beans, green and yellow vegetables, light-colored vegetables, fruits, mushrooms, seaweeds, seafoods, meats, eggs, milk and dairy products, and alcoholic beverages) were calculated by Education Software Co., Ltd. (Tokyo, Japan).

### 2.3. Collection and Preparation of Stool Samples and DNA Extraction

Each subject was given a stool collection tube containing 2 mL of RNAlater (Thermo Fisher Scientific, Waltham, MA, USA) and zirconia beads, which had been weighed beforehand. Subjects introduced approximately 0.5 g of fresh stool into the tube with a stool collection spoon and then shook the sealed tube thoroughly. The tubes were triple-wrapped and transported to KSO. On receipt, samples were anonymized and immediately stored at 4 °C. Within a week they were transported to Yakult Central Institute and stored at 4 °C until preparation for analysis.

Stool samples were prepared as described by Kubota et al. [17]. Each sample was weighed and diluted 1:9 with RNAlater. The mixture was suspended for 10 min at 1048 rpm on a ShakeMaster Auto (Bio Medical Science Co., Ltd., Tokyo, Japan). Then 200 µL of the suspension was introduced into a 2-mL screw-cap tube containing 1 mL of phosphate-buffered saline (PBS). The contents were stirred on a vortex mixer and centrifuged at 13,000 × g for 5 min, and 1 mL of supernatant was discarded. This process was performed a second time. The resulting 200 µL of suspension was stored at −30 °C until DNA extraction. DNA was extracted as described by Matsuki et al. [18].

### 2.4. Quantification of Total Bacteria by Quantitative PCR

Total bacterial counts in stool sample were quantified according to the method described by Shima et al. [15]. The primers used were UniF (5′-GTGSTGCAYGGTCGTCA-3′) and UniR (5′-ACGTCRTCCMCNCCTTCCTC-3′) [19]. The standard curve was obtained by amplifying DNA extracted from *Faecalibacterium prausnitzii* ATCC27768^T^ with the UniF and UniR primers.

PCR was performed on an ABI PRISM 7900HT Fast Real-Time PCR System (Thermo Fisher Scientific). The PCR reaction solution (10 µL) contained 10 × PCR Mg^2+^-free buffer (Takara Bio Inc., Shiga, Japan) (1.0 μL), 25 mM MgCl_2_ solution (1.0 μL), dNTP mixture (2.5 mM each) (0.8 μL), 1:300-diluted SYBR Green I (0.03 μL), 50 × ROX Reference Dye (0.2 μL), 5 units/μL TaKaRa Taq (Takara Bio Inc.) (0.04 μL), 1.1 µg/μL Taq Start antibody (0.05 μL), 10 µM each of primers (0.2 μL), Nuclease-Free Water (1.48 μL), and template DNA (5.0 μL). The PCR conditions were an initial 95 °C for 5 min, followed by 40 cycles of 94 °C for 20 s, 55 °C for 20 s, and 72 °C for 50 s. The detection limit of bacterial counts was 10^6^ cells/g feces.

### 2.5. Amplification of the 16S rRNA Gene Region and Next-Generation Sequencing

DNA was prepared for 16S rRNA gene amplicon sequencing according to the Earth Microbiome Project (EMP) protocol [20,21]. Concentrations of DNA extracted from stool samples were measured with a NanoDrop 2000c (Thermo Fisher Scientific) and diluted to 10 ng/µL. The V1–V2 region of the 16S rRNA gene in diluted DNA samples was amplified using the 27Fmod2 forward primer (5′-AATGATACGGCGACCACCGAGATCTACACTCTTTCCCTACACGACGCTCTTCCGATCTAGRGTTYGATYMTGGCTCAG-3′) and the 338R reverse primer which contains Golay barcode and illumina adapter sequences (5′-CAAGCAGAAGACGGCATACGAGAT-NNNNNNNNNNNN-GTGACTGGAGTTCAGACGTGTGCTCTTCCGATCTGCTGCCWCCCGTAGGWGT-3′) [22]. PCR was performed on an ABI 7500 Real-Time PCR System (Thermo Fischer Scientific). The PCR reaction solution (50 μL) contained 2 × TB Green Premix ExTaq (Tli RnaseH plus) (Takara Bio Inc.) (25 μL), Nuclease-Free Water (22 μL), 100 nM each of primers (1 μL), and template DNA (1 μL). The PCR conditions were an initial 95 °C for 30 s, followed by 95 °C for 5 s, 55 °C for 30 s, and 72 °C for 40 s. According to the previous studies [22,23], PCR was monitored by SYBR signal, and stopped before signal saturation to minimize bias and erroneous product. The PCR products were purified with an AMPure XP Kit (Beckman Coulter, Brea, CA, USA) and quantified with a Quant-iT PicoGreen dsDNA Kit (Invitrogen, Leek, the Netherlands). The library was constructed by mixing equal amounts of DNA for every sample, and sequenced on a MiSeq System (Illumina, San Diego, CA, USA) using a paired-end 2 × 250 bp cycle run and MiSeq Reagent Kits v2 (Illumina). As a result, 15,722,623 amplicon sequence reads were obtained (from 10,349 to 119,158 reads per sample). The raw sequence paired-end files are deposited in the DDBJ databases with the accession number PRJDB9936.

### 2.6. Processing of 16S rRNA Gene Sequence Data

The amplicon sequence reads were processed through the QIIME 2 (ver. 2018.8, https://qiime2.org/) [24]. Raw amplicon sequencing data is processed into the table of exact amplicon sequence variants (ASVs) present in each sample using the DADA2 plugin [25]; phix reads and chimeric sequences were filtered using pooled consensus method. The resulting feature table was used for taxonomic assignment based on the Greengenes reference database (ver. 13.8) [26] by training a Naïve Bayes Classifier using the q2-feature-classifier plugin. Absolute count data form taxonomic assignment was normalized into relative abundances at phylum and genus level using the taxa plugin. De novo multiple alignment was performed using the MAFFT method [27] and filtered to remove highly variable regions. Phylogenic trees were constructed using the FastTree2 and Midpoint Root methods [28,29]. The α-diversity of samples were analyzed at a sampling depth of 10,000, allowing retention of 354 samples. Shannon’s diversity, observed ASVs, and Faith’s phylogenetic diversity indexes were calculated by QIIME2 using the diversity plugin for analyzing α-diversity.

### 2.7. Estimation of the Number of Gut Bacteria

The number of gut bacteria at the genus level in each subject was calculated by multiplying the total bacterial count determined by quantitative PCR by the compositional data at the genus level obtained by 16S rRNA gene amplicon analysis.

### 2.8. Stastical Analysis

Statistical analysis was performed in R 3.5.0 software. The Fischer’s exact test was used for the analysis of categorical variables (“fisher.test” function, “stats” package). The correlation between α-diversity indexes (Shannon’s diversity, observed ASVs, and Faith’s phylogenetic diversity [PD]) and intake of each food group was assessed by Spearman’s partial correlation coefficient (“pcor.test” function, “ppcor” package), using sex, age, BMI, total calorie intake, and presence/absence of smoking as covariates. The association between dissimilarity (β-diversity) of microbiota and intake of each food group was analyzed by permutational ANOVA (PERMANOVA) with 10,000 permutations to calculate *p*-values (“adonis” function, “vegan” package), using sex, age, BMI, and total energy intake as covariates. Bray–Curtis and Jaccard metrics, as β-diversity indexes, were calculated by the “vegdist” function (“vegan“package) using the number of gut bacteria at the genus level obtained as described above. The association between the number of gut bacteria and habitual dietary intake was analyzed by multiple linear regression analysis (“lm” function, “stats” package). The number of each gut bacteria at genus level was used separately as an objective variable, while the intake of each food group as an explanatory variable and sex, age, BMI, and total energy intake as covariates, were keeping same for each equation. The bacteria targeted for analysis were defined as those detected in ≥20% of subjects. The number of gut bacteria was log-transformed for analysis. Values less than the detection limit (10^6^ cells/g feces) were replaced with half of the detection limit. The significance level was set at *p* < 0.05 in all statistical analyses.

## 3. Results

### 3.1. Characteristics of Subjects

The characteristics of the 354 subjects are shown in Table 1. The median age was 40 years (interquartile range (IQR): 29–50 years) and the median BMI was 21.9 kg/m^2^ (IQR: 20.0–24.0 kg/m^2^). Former smokers comprised 9.6% and current smokers 15.3%.

The habitual intake of each food group is shown in Table 2. The mean daily energy intake was 8537.5 ± 1069.4 kJ/day (mean ± SD). Intake of grains was the highest (381.6 ± 95.6 g/day, mean ± SD) among the food groups assessed.

The total gut bacterial count was 10.8 ± 0.3 log_10_ cells/g feces (mean ± SD) (Figure 1A). 16S rRNA gene amplicon analysis detected 215 genera in 15 phyla. The most commonly identified bacteria at the phylum level were the Bacteroidetes (abundance: 48.8% ± 14.2%, mean ± SD) and the Firmicutes (37.2% ± 10.9%), followed by the Actinobacteria (11.4% ± 11.1%), the Proteobacteria (1.73% ± 2.02%), and the Verrucomicrobia (0.198% ± 0.663%) (Figure 1B). The composition at the genus level is shown in Appendix A.

### 3.2. Association between α-Diversity of Gut Microbiota and Dietary Food Intake

Spearman’s partial correlation coefficient showed that mushroom intake was positively correlated with observed ASVs (ρ = 0.114) and Faith’s PD (ρ = 0.126) (Figure 2). No intake of foods was significantly correlated with Shannon’s diversity index. Intake of grains was significantly negatively correlated with Faith’s PD (ρ = −0.132), and intake of beans was significantly positively correlated with observed ASVs (ρ = 0.102) (Figure 2).

### 3.3. Association between β-Diversity of Gut Microbiota and Dietary Food Intake

In PERMANOVA, the Bray–Curtis and Jaccard indexes were significantly associated with the intake of fruits, mushrooms, seaweeds, seafoods, and alcoholic beverages (Figure 3).

### 3.4. Associations between Number of Gut Bacteria and Food Intake Related to α-Diversity of Gut Microbiota

In multiple regression analysis of food groups related to α-diversity of gut microbiota and number of gut bacteria, intake of mushrooms (also related to β-diversity) was associated negatively with the number of *Parabacteroides* (Table 3). Intake of grains was correlated positively with numbers of *Bacteroides*, *Streptococcus*, and *Veillonella* and negatively with numbers of *Lactobacillus* and *Lactococcus*. Intake of beans was correlated positively with numbers of *Prevotella*, *Bacillus*, *Clostridium*, *Roseburia*, *Faecalibacterium*, *Ruminococcus*, and *Meganomas* and negatively with numbers of *Eubacterium* and *Fusobacterium*.

### 3.5. Associations between Number of Gut Bacteria and Food Intake Related to β-Diversity of Gut Microbiota

In multiple linear regression analysis of food groups related to β-diversity of gut microbiota and number of gut bacteria, intake of fruits was associated positively with numbers of Streptococcus and *Butyricicoccus* and negatively with the number of *Alistipes* (Table 4). Intake of seaweeds was correlated positively with the number of *Subdoligranulum* and negatively with the number of *Streptococcus*. Intake of seafoods was correlated negatively with the number of *Bacteroides*. Intake of alcoholic beverages was correlated positively with the number of *Fusobacterium* and negatively with numbers of *Actinomyces* and *Clostridium*.

## 4. Discussion

16S rRNA gene amplicon analysis by next-generation sequencing is used generally to analyze the relationship between diet and gut microbiota. This powerful technique allows comprehensive identification of gut microbiota composition. However, although it can only provide composition data, it cannot quantify gut microbial counts. Recent studies have pointed out that quantitative profiling of gut microbiota is necessary to accurately understand the association of gut microbiota with the intestinal environment and the host [30,31]. Our analysis using data of gut microbiota composition obtained from 16S rRNA gene amplicon analysis and the total number of gut bacteria obtained from quantitative PCR based on the targeting of bacterial 16S rRNA consensus sequence enabled us to reveal in detail the association between habitual dietary intake and gut microbiota.

We found a negative correlation between intake of grains and α-diversity of gut microbiota (Figure 2). A large-scale cohort study in 1106 Belgians showed a negative correlation between intake of bread and α-diversity of gut microbiota [32]. Among grains, rice and bread are major staple foods in Japan and account for a large proportion of energy consumption [33]. These results, taken together, indicate that intake of bread, at least, is most likely to be negatively associated with α-diversity of gut microbiota. We found a positive correlation between intake of beans and mushrooms and α-diversity (Figure 2). Although white button mushrooms increased α-diversity of gut microbiota in mice [34], our study is the first to our knowledge that shows such an association in humans. Future studies are needed to reveal whether these associations are typical in Japanese people. Intake of milk and dairy products negatively correlated with Shannon’s diversity index, while positively correlated with Observed ASVs and Faith’s PD (Figure 2). It is unclear why intake of milk and dairy products had the opposite effect to α-diversity, and additional study to confirm their relationship was needed.

Significant associations of the intake of fruits and alcoholic beverages with β-diversity of gut microbiota were reported in the Belgian study (*n* = 1106) [32] and in a Dutch study (*n* = 1135) [35]. These results are consistent with ours (Figure 3). On the other hand, an association of the intake of mushrooms, seaweeds, and seafoods with β-diversity was not reported. Although there are no reports of international comparisons in mushroom intake between Japan and other countries, intake of seaweeds is higher in Japan than in Western countries [36] and consumption of seafoods is greater in Japan than in EU countries [37]. From these reports, the associations between β-diversity of gut microbiota and these food groups newly observed in this study is likely to be attributable to the genetic background of the subjects and cultural differences in diet.

Grains contain an abundance of dietary fibers such as cellulose and hemicellulose. Resistant polysaccharides such as dietary fibers that reach the gastrointestinal tract are metabolized by a variety of bacteria, including *Bacteroides*, *Bifidobacterium*, *Clostridium*, *Eubacterium*, *Lactobacillus*, *Ruminococcus*, and *Roseburia* [38]. Intervention of a diet rich in resistant polysaccharides increased the abundance of *Bacteroides* [39], consistent with our results (Table 3). In other report, a diet rich in resistant polysaccharides also increased the number of gut bacteria such as bifidobacteria and lactobacilli [40]. However, we found no association between the intake of grains and the number of *Bifidobacterium*. In addition, we found a negative association with the number of *Lactobacillus* (Table 3), contrary to the previous study [40]. This difference may be because we analyzed the effect of a habitual diet, not intervention of specific diet. As the previous study used fluorescence in situ hybridization (FISH) to analyze gut microbiota [40], the difference in methods of analysis is likely to have also affected the results. We found new associations of the intake of grains with the numbers of *Lactococcus* and *Streptococcus* (Table 3), not previously reported. The use of the numbers of gut bacteria for analysis has made it possible to find such associations for the first time, allowing their causal relations to be elucidated.

Beans, like grains, also contain an abundance of dietary fibers, more so than in other foods. Heating beans leads to the creation of resistant starch and increases the content of dietary fibers. Bacterial species such as *Bifidobacterium adolescentis*, *Ruminococcus bromii*, and *Eubacterium rectale* are involved in the digestion of resistant starch [41,42]. A diet rich in resistant starch increases the abundance of *Ruminococcus*, *Roseburia*, and *Eubacterium* [43]. We found a positive association between intake of beans and numbers of *Roseburia* and *Ruminococcus* (Table 3), consistent with a previous study [43], but a negative association with the number of *Eubacterium* (Table 3), inconsistent with a previous study [41]. Although the intake of resistant polysaccharides is suggested to increase the number of total gut bacteria [44], microbial analysis by 16S rRNA gene amplicon sequencing does not reflect an increase in the number of total bacteria. Therefore, the abundance of *E. rectale* in a previous study [41] may not have reflected actual enteric conditions. The intake of beans was positively associated with the number of *Bacillus* (Table 3), possibly because natto (fermented soybeans), a typical Japanese food, is rich in live *Bacillus subtilis*, and intake of natto may have influenced the number of *Bacillus* among gut bacteria, speculatively.

Mushrooms are rich in β-glucans, and De Angelis et al. reported that a β-glucan-rich diet increased the abundance of *Clostridium*, *Roseburia*, and *Ruminococcus* and decreased that of *Faecalibacterium* and *Fusobacterium* [45]. However, we found no association between intake of mushrooms and the numbers of these genera (Table 3). Hess et al. also reported that an intervention of mushrooms in healthy adults in the USA increased the abundance of Bacteroidetes and decreased that of Firmicutes [46]. In our study, however, we found a negative association between intake of mushrooms and the number of *Parabacteroides* (belong to Bacteroidetes) (Table 3), inconsistent with previous study [46]. We do not know whether this discrepancy is due to differences in ethnicity and region or in the methods of gut microbiota analysis. Further studies are needed.

A three-day diet of vegetable and fruit juice rich in polyphenols, oligosaccharides, and dietary fiber in US residents decreased the abundance of Bacteroidetes and increased that of Firmicutes [47]. Although details about the fruits used in the study are unknown, we also found a negative association of fruit intake with the number of *Alistipes* (belong to Bacteroidetes) and a positive association with the numbers of *Streptococcus* and *Butyricococcus* (belong to Firmicutes) (Table 4). Therefore, nutrients in fruits, such as polyphenols, oligosaccharides, and dietary fibers, are suggested to affect numbers of gut bacteria, regardless of ethnicity and region.

Seaweeds are rich in polysaccharides such as alginates, fucoidans, and sulfated galactans [48]. Although there are no studies in humans of the effect of seaweeds and polysaccharides in them on gut microbiota, a study in rats showed that intake of *Laminaria japonica* increased the abundance of lactic acid-producing bacteria such as *Subdoligranulum* and *Streptococcus* [49]. *Subdoligranulum* reportedly assimilates a wide range of carbohydrates and uses enzymes such as β-glucosidase and β-galactosidase in their degradation of carbohydrates [50]. No studies have reported on the assimilation of polysaccharides by *Subdoligranulum*, but we found a positive association between intake of seaweeds and the number of *Subdoligranulum* (Table 4). Therefore, polysaccharides in seaweeds are likely to affect the number of *Subdoligranulum*. On the other hand, we found a negative association between intake of seaweeds and the number of *Streptococcus* (Table 4), probably due to the difference in the species targeted.

Few studies have directly analyzed the association between intake of seafoods and gut microbiota. Most studies have focused on unsaturated fatty acids, which are abundant in seafoods. Urwin et al. reported that intake of salmon rich in mono- and polyunsaturated fatty acids did not affect gut microbiota in pregnant women [51]. The analysis of gut microbiota using FISH by Fava et al. revealed that a diet rich in monounsaturated fatty acids did not affect the abundance but decreased the total number of gut bacteria [52]. Although we found no significant association between intake of seafoods and the total number of bacteria in this study, to our knowledge, this is the first study to show a negative association with *Bacteroides* in healthy adults (Table 4). This association may have been revealed not by conventional analysis based on the abundance of gut bacteria, but by analysis based on their number.

Intake of alcohol is widely known to affect the composition of gut microbiota [53]. Intake of alcoholic beverages was associated negatively with the number of *Clostridium* and positively with that of *Fusobacterium* (Table 4). The abundance of clostridia decreases in alcoholic patients [54] and that of the Fusobacteriaceae is high in patients with alcoholic liver cirrhosis [55]. Our study may be the first to show an association of alcohol consumption with *Clostridium* and *Fusobacterium* in healthy adults. Daily alcohol consumption can be a risk factor for colorectal cancer [56,57]. *Fusobacterium nucleatum* is involved in the development and progression of colorectal cancer [58,59]. Therefore, there may be an association between increased *Fusobacterium* due to alcohol consumption and the development of colorectal cancer.

Our study had several limitations. First, we conducted a single questionnaire survey on antibiotic use within one week before stool collection. Therefore, we did not exclude the impact of taking antibiotics prior to one week. Second, we used an FFQ to assess dietary intake; however, as a method to assess habitual long-term dietary intake, it cannot provide information on an unbalanced daily diet. A short-term dietary intervention can influence gut microbiota [11]; therefore, diet just before stool collection may have a great effect on the results of gut microbiota analysis. To clarify the causal relationship between dietary intake and the number of gut bacteria, interactions between each food items should be taken into consideration. A long-term subsequent analysis of the association between diet and gut microbiota is thus necessary to reveal the detailed causal relationship of them. Third, we obtained the number of gut bacteria by multiplying the total bacterial count calculated by quantitative PCR by the bacterial composition data at the genus level from 16S rRNA gene amplicon analysis. However, the number of gut bacteria was likely to be substantially more or less than the actual number because the number of 16S rRNA gene copies differed by bacterial species. In addition, 16S rRNA gene amplicon analysis can measure only dominant gut bacteria; therefore, the effect of diet on bacteria that are small in number but affect the host’s health cannot be evaluated. However, in contrast to the data based on the abundance of bacteria obtained by decoding 16S rRNA gene amplicon gene sequences, this method enables comprehensive quantitative analysis of gut microbiota. This has thus made it possible to quantify the association between habitual dietary intake and gut microbiota.

## 5. Conclusions

We revealed the association between habitual intake of various diets and specific gut bacteria in healthy Japanese adults. The results may not only help us to understand the complex relationship between habitual diet and gut microbiota, but also provide important information on the regional specificity of gut microbiota. Specific dietary interventions could be expected to help establish a method to induce gut microbiota beneficial to the host’s health.

## Figures and Tables

**Figure 1 nutrients-12-02414-f001:**
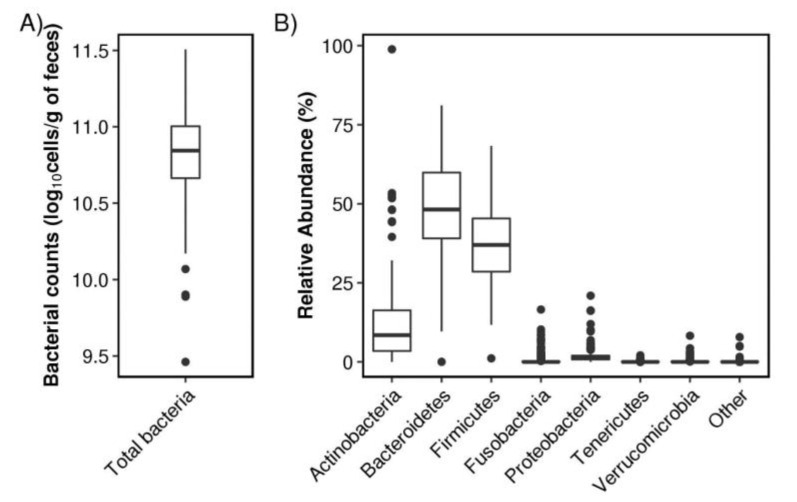
Gut microbiota of the subjects in this study. (**A**) Total gut bacterial counts. (**B**) Abundance of bacteria at the phylum level.

**Figure 2 nutrients-12-02414-f002:**
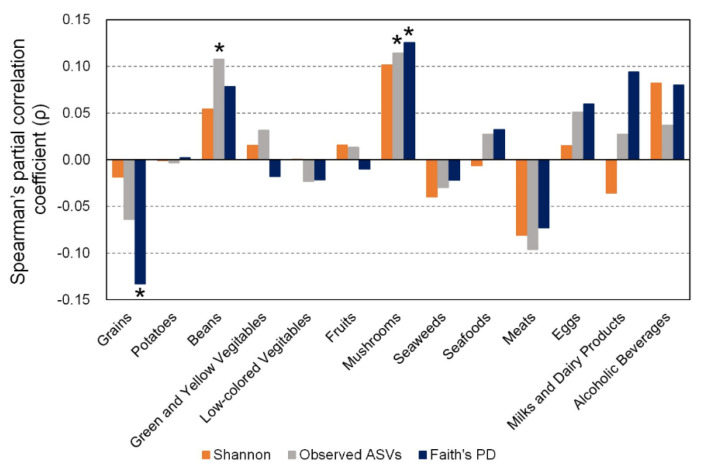
Association between α-diversity of gut microbiota and intake of each food group. Spearman’s partial correlation coefficient adjusted for sex, age, BMI, smoking habit, and total energy intake was calculated for α-diversity indexes (Shannon index, observed ASVs, and Faith’s PD) and intake of each food group. * *p* < 0.05.

**Figure 3 nutrients-12-02414-f003:**
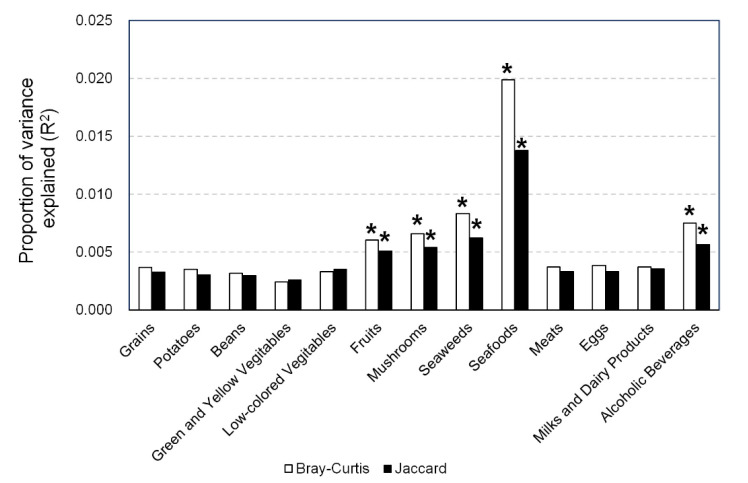
Association between β-diversity of gut microbiota and intake of each food group. The proportion of variance explained (*R*^2^) of each food group was calculated by analyzing the association of β-diversity indexes (Bray–Curtis and Jaccard) and dietary intake using permutational ANOVA (PERMANOVA) adjusted for sex, age, BMI, smoking habit, and total energy intake. * *p* < 0.05.

**Table 1 nutrients-12-02414-t001:** Characteristics of the subjects in this study.

Characteristic	Median (IQR)	*n* (%)	*p* Value
Sex			
Female		173 (48.9)	0.60
Male		181 (51.1)	
Age (years)	40 (29–50)		
20–29		89 (25.1)	0.96
30–39		85 (24.0)	
40–49		89 (25.1)	
50–59		91 (25.7)	
BMI (kg/m^2^)	21.9 (20.0–24.0)		
<18.5		33 (9.3)	<0.001
18.5–<25.0		258 (72.9)	
≥25.0		63 (17.8)	
Smoking status			
Never		266 (75.1)	<0.001
Former		34 (9.6)	
Current		54 (15.3)	

*p* values were estimated by Fisher’s exact test. IQR, interquartile range; BMI, body mass index.

**Table 2 nutrients-12-02414-t002:** Habitual dietary food intake of the subjects in this study.

Food Intake	
Total energy (kJ/day)	8537.5 ± 1069.4
Food group (g/day)	
Grains	381.6 ± 95.6
Potatoes	41.2 ± 8.7
Beans	77.3 ± 46.3
Green and yellow vegetables	162.9 ± 50.2
Light-colored vegetables	181.0 ± 36.6
Fruits	148.1 ± 27.1
Mushrooms	16.5 ± 11.7
Seaweeds	10.8 ± 5.3
Seafoods	92.8 ± 3.1
Meats	76.7 ± 20.0
Eggs	35.4 ± 8.6
Milks and dairy products	192.5 ± 85.0
Alcoholic beverage	139.3 ± 195.3

Data was displayed in Mean ± SD.

**Table 3 nutrients-12-02414-t003:** Significant associations between food groups associated with α-diversity and numbers of gut bacteria at the genus level.

Food Group		Unstandardized Coefficients		Standardized Coefficients	*p* Value
		B	Std. Error	β	
Grains	Bacteroidetes				
	*Bacteroides*	0.000953	0.000401	0.190	0.018
	Firmicutes				
	*Lactobacillus*	−0.00410	0.00207	−0.161	0.048
	*Lactococcus*	−0.00414	0.00158	−0.211	0.0093
	*Streptococcus*	0.00182	0.000846	0.173	0.032
	*Veillonella*	0.00482	0.00209	0.188	0.022
Beans	Bacteroidetes				
	*Prevotella*	0.00852	0.00401	0.120	0.034
	Firmicutes				
	*Bacillus*	0.00991	0.00284	0.201	<0.001
	*Clostridium*	0.00630	0.00270	0.133	0.020
	*Roseburia*	0.00670	0.00295	0.130	0.024
	*Faecalibacterium*	0.00529	0.00207	0.143	0.011
	*Ruminococcus*	0.00473	0.00197	0.136	0.017
	*Meganomas*	0.00716	0.00335	0.122	0.034
	*Eubacterium*	−0.00757	0.00249	−0.176	0.0025
	Fusobacteria				
	*Fusobacterium*	−0.00912	0.00313	−0.162	0.0039
Mushrooms	Bacteroidetes				
	*Parabacteroides*	−0.0323	0.00675	−0.261	<0.001

**Table 4 nutrients-12-02414-t004:** Significant associations between food groups associated with β-diversity and numbers of gut bacteria at the genus level.

Food Group		Unstandardized Coefficients		Standardized Coefficients	*p* Value
		B	Std. Error	β	
Fruits	Bacteroidetes				
	*Alistipes*	−0.0132	0.00528	−0.163	0.013
	Firmicutes				
	*Streptococcus*	0.00519	0.00245	0.139	0.035
	*Butyricicoccus*	0.00982	0.00492	0.134	0.047
Seaweeds	Firmicutes				
	*Streptococcus*	−0.0440	0.0208	−0.137	0.035
	*Subdoligranulum*	0.125	0.0545	0.151	0.022
Seafoods	Bacteroidetes				
	*Bacteroides*	−0.0111	0.00355	−0.309	0.0020
Alcoholic beverage	Actinobacteria				
	*Actinomyces*	−0.00128	0.000606	−0.121	0.036
	Firmicutes				
	*Clostridium*	−0.00145	0.000629	−0.129	0.022
	Fusobacteria				
	*Fusobacterium*	0.00170	0.000734	0.127	0.021

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
