# Peer review of "Impacts of Habitual Diets Intake on Gut Microbial Counts in Healthy Japanese Adults"

_nutrients, 2020, doi:10.3390/nu12082414_

Round 1

Reviewer 1 Report

The reviewed work is a unique study which presents data related to impacts of habitual Japanese diets on gut microbial counts. According to my knowledge there is no similar paper treating about this subject. The study is well designed, includes significant number of subjects, data are appropriately presented. Discussion is clear and coherent.

Despite the weaknesses mentioned by authors this study article could be interested for most readers.

Author Response

We thank you very much for your kind comment.

Reviewer 2 Report

My comments for the authors MDPI manuscript

  • Did the authors determined ‘OTUs’ or ‘ASV’s from the data? The DADA2 algorithm calculates the ‘ASVs’ or ‘ESVs’ and not the conventional ‘OTUs’. All the results shows OTUs which is little confusing. Can the authors clarify it?
  • Line 140: spelling mistake: ‘phynum’ or ‘phylum’
  • Line 143: It is not clear if the alpha diversity, OTUs and phylogenetic diversity indexes were calculated in Qiime or any R package or other tools. Also, please cite the appropriate R packages wherever used.
  • How do the authors interpret the situation at Figure 2 where the ‘Milk and dairy products’ show negative partial correlation for shannon but positive for ‘observed OTUs’ and ‘Faith’s PD’?
  • It is not clear how the authors performed the multiple linear regression (using ‘lm’). Did they used microbiome (which level, phylum or genus etc) as the outcome? Did they used each bacterial count as the outcome separately for all the bacteria while keeping the predictors same for each equation? This needs to be clear.
  • It seems that the food frequency was collected for the last one year and the stool sample was collected once at the end of the study. It is possible that the volunteers might have changed their diet in the last few months (say 2-3 months). In that case the microbiome may be a better representative of the recent diet rather than the whole year. What is the opinion of authors on it?
  • Spelling mistake in Figure 2. It should be ‘potatoes’.
  • The authors have excluded those individuals who have taken antibiotics within 1 week of stool collection (line 72). Isn’t one week to less to consider the effect of antibiotic? The effect of it can be visible even after one month and sometimes up to six months!!

Reviewer 3 Report

In this work, Sugimoto et al. present the relationship between habitual dietary intake and gut microbiota composition in 354 healthy Japanese adults. Although several other studies already presented similar results, this work is the first to focus on Japanese diet, to the best of my knowledge.

The paper is well-written, and the results are clearly described and contextualized. Although the novelty of the work is not high, in my opinion, the results here presented could be of interest for readers working in nutrition and microbiology fields. Only few minor issues should be addressed by the authors.

  • Table 1: the values should be better expressed as median values (IQR). Furthermore, p-values of the comparisons should be indicated.
  • Figures 2 and 3: please correct the name of the foods (e.g.: “Yello vegitables”)
  • Alcohol intake correlated to diversity: was the average intake of alcohol calculated?
  • Lines 255-256: the sentence is repeated, delete it.
  • Lines 283-85: this sentence is speculative, as you did not assess the content of B. subtilis in the food consumed.
